# Human-guided Rule Learning for ICU Readmission Risk Analysis

Lincen Yang
l.yang@liacs.leidenuniv.nl
Leiden University
Leiden, The Netherlands

Matthijs van Leeuwen
m.van.leeuwen@liacs.leidenuniv.nl
Leiden University
Leiden, The Netherlands

## ABSTRACT

Interactive machine learning systems that can incorporate human feedback for automatic model updating have great potential use in critical areas such as health care, as they can combine the strength of data-driven modeling and prior knowledge from domain experts. Designing such a system is a challenging task because it must enable mutual understanding between humans and computers, relying on interpretable and comprehensible models. Specifically, we consider the problem of incorporating human feedback for model updating in rule set learning for the task of predicting readmission risks for ICU patients. Building upon the recently proposed Truly Unordered Rule Sets (TURS) model, we propose a certain format for feedback for rules, together with an automatic model updating scheme. We conduct a pilot study and demonstrate that the rules obtained by updating the TURS model learned from ICU patients' data can empirically incorporate human feedback without sacrificing predictive performance. Notably, the updated model can exclude conditions of rules that ICU physicians consider clinically irrelevant, and thus enhance the trust of physicians.

## CCS CONCEPTS

• **Computing methodologies → Rule learning**.

## KEYWORDS

Interactive machine learning, Probabilistic rules, Human-in-the-loop, Machine learning for healthcare

**ACM Reference Format:**
Lincen Yang and Matthijs van Leeuwen. 2024. Human-guided Rule Learning for ICU Readmission Risk Analysis. In *Proceedings of Make sure to enter the correct conference title from your rights confirmation email (Conference acronym 'XX)*. ACM, New York, NY, USA, 6 pages. https://doi.org/XXXXXXX.XXXXXXX

## 1 INTRODUCTION

In critical areas such as health care, developing machine learning models that domain experts can comprehend and trust potentially has great societal impact. Specifically, in intensive care units (ICU) where patients are monitored intensively, patients conditions are to a large extent recorded digitally, which provides the foundations for building decision support systems with data-driven models [4].

We consider the problem of predicting the probability of readmission to the ICU within a short period (7 days) after a patient is discharged from the ICU and moved to a normal ward. Such readmission risk for patients is clinically relevant, as it is observed that patients who are readmitted often become much worse in comparison to their condition when they were in the ICU previously [9, 19]. Thus, the readmission itself is a key factor that is highly correlated with the patient's condition; as a result, predicting the readmission risk can both facilitate efficient ICU resource management and prevent discharging patients improperly. In practice, beds in the ICU are a very scarce and costly resource; thus, discharging patients from the ICU smartly can help distribute the resource to patients who need it more.

As physicians are responsible for estimating the risk of discharging a patient from the ICU, data-driven models only brings benefits if physicians trust the model and are willing to use it in practice. To build trust, the data-driven model needs to have interpretability for domain experts to comprehend what is going on [11]. Further, beyond interpretability, the situation when physicians and machine learning models disagree must be properly handled [7, 13, 15]. That is, when the model gives a probabilistic prediction together with explanations, what if the physician disagrees with the prediction and/or the explanation? For instance, the model could identify a factor that is known to be irrelevant clinically as important for predicting readmission risk for a single patient. In this situation, it would be ideal if the physician would give this feedback to the machine learning model; further, if the model can be automatically updated when receiving the feedback from human, the physician could trust the model next time when the model gives the same explanation and prediction for a similar patient in the future.

Thus, interaction between humans (i.e., physicians in the ICU in this case) and the machine learning model is crucial in such a scenario, which requires the human to understand the machine, and at the same time, the machine to understand the human.

As probabilistic rules [6] are directly readable by humans, rule-based models are in principle comprehensible to humans [12]. However, traditional probabilistic rules raise the challenge for human-guided rule learning, in the sense that rules cannot be modified (in a data-driven way) without affecting other "overlapping" rules, in which two rules being overlapped is defined as the situation when some instances satisfy the conditions of both rules. This motivates us to adopt the recently proposed Truly Unordered Rule Set (TURS) models [21]. In Section 3, we will briefly review probabilistic rules, discuss further the issue caused by overlaps, and describe the TURS model as preliminaries.

Building upon TURS, the challenges remain unresolved that 1) how and in what formats the feedback from domain experts can be incorporated, and 2) how rule-based models can be updated according to human feedback. To tackle these challenges, we introduce

a human-guided rule updating scheme based on the TURS model. Specifically, we present a rule set model to a human user, and ask which rules they dislike and *why*. While it seems tempting to allow the user to specify their reasons in natural languages, this cannot guarantee the *transparency* of the model updating process. Note that automatically updating the model based on human feedback has now become part of the machine learning system, which we aim to make interpretable altogether.

Thus, we constrain the feedback in certain formats, propose a transparent human-guided model updating scheme, and conduct an empirical pilot study by applying our method to a dataset collected at the ICU of Leiden University Medical Center (LUMC) in the year 2020. To this end, we ask a domain expert from LUMC to identify rules with clinically irrelevant variables. Our results demonstrate that with human-guided rule learning, probabilistic rules can be updated to meet users' preferences without sacrificing the predictive performance of the model. To the best of our knowledge, we are the first to develop a human-guided machine learning system based on probabilistic rules.

## 2 RELATED WORK.

Involving humans in the loop in machine learning systems has been studied extensively in computer visions and natural language process [5]. However, for text and image datasets, data point makes more sense to humans by themselves than those in tabular datasets— the data type for our task—unless the tabular data has a very low dimensionality. Further, their goals are often to incorporate humans' prior knowledge to increase the accuracy, while our goal is to make the model more trustworthy to domain experts. On the other hand, several methods exist that allow user to influence the learned model. For instance, Ware et al. [18] proposed to directly build classifiers (decision trees) with the help of visualizations. Kapoor et al. [8] allow users to update the model by refining the confusion matrix. Finally, other works include involving humans in the loop for feature engineering [2] and data labelling [1]. Although these methods provide a certain degree of control to humans, our work is different in the following aspects: 1) we let users specifying the disliked variables and eliminating such variables via local model updating instead of re-training the whole model, 2) we build the model on a rule set that summarizes the original data as comprehensible patterns, resolving the issue that each single data point may be hard to perceive for humans, and 3) we specifically focus on the critical and sensitive application task in the healthcare domain, with the main goal as enhancing the trust between humans and machine learning systems.

## 3 TRULY UNORDERED RULE SETS

We first review the definition of probabilistic rules, and then discuss the truly unordered rule set (TURS) model.

### 3.1 Probabilistic rules

Denoting the feature as $X = (X_1, ..., X_m)$ and the target variable as $Y$, a probabilistic rule is in the form of *"IF $X$ meets certain conditions, THEN $P(Y) = \hat{P}(Y)$"*, where the *condition* of the rule is a conjunction of literals (i.e., connected by the logical AND). Each literal takes the form of "$X_j \geq$ (or $<$) $v_j$" for some value $v_j$ if $X_j$ is continuous, or

$X_j = v_j$ if $X_j$ is categorical. Further, $\hat{P}$ denotes the class probability estimator for this rule. When a instance satisfies the condition of a rule, we refer to the instance as being *covered* by this rule.

### 3.2 Rule-based models

While a single rule describes a subset of data only, a global model can be formed by putting a set of rules together, as a rule list [3, 20] or an unordered rule set [10, 22]. In a rule list, rules are connected by the "IF" and (multiple) "ELSE IF" statements (e.g., IF condition A, Probability of readmission is 0.1; ELSE IF condition B, Probability of readmission is 0.4). Rule lists are hard to comprehend as the condition of a single rule depends on all its preceding rules.

Further, in unordered rule sets, rules are claimed to be unordered whereas implicit orders are usually imposed, as pointed out by Yang and van Leeuwen [21]. When an instance satisfies the conditions of multiple rules, these rules are often ranked based on their accuracy [22] or F1-score [10]. Then, the higher-ranked rule is used for predicting that single instance covered by multiple rules, while the lower-ranked rule is disregarded. However, these implicit ranks among rules cause issues when humans want to intervene by providing feedback to the rules (e.g., they like/dislike certain variables), and let the rules be automatically updated. This is because rules become entangled due to the existence of ranks; as a result, single rules cannot be re-trained without affecting other rules. Further, with implicit orders, the condition of a single rule also depends on other higher-ranked rules; thus, similar to rule lists, comprehending a single rule requires checking all higher-ranked rules.

### 3.3 The TURS model

The TURS model eliminates both implicit and explicit orders among rules by formalizing a set of rules as a global probabilistic model in a novel way. Specifically, when two rules overlap, the instances that satisfy the conditions of both rules are modelled to express "uncertainty", in the sense that the TURS model is uncertain which rule can "better" describe these instances. Intuitively, this can happen when 1) the overlap contains very few data points, and/or 2) the (empirical) class probabilities for instances contained in the overlap is "similar" to either rule. According to the Occam's razor principle, creating a separate rule to cover exactly these instances contained in the overlap is not preferred, as the gain for the model's goodness-of-fit is little in comparison to the increase of model complexity, which in practice may lead to overfitting [21].

Particularly, learning a TURS model from data has been formalized as a task of model selection based on the minimum description length (MDL) principle [14], in which the MDL principle is a formalization of Occam's razor.

The TURS model paves the way towards an interactive rule learning process with the following two advantages over existing methods for learning rule lists and rule sets, in which rules are respectively explicitly and implicitly ordered.

The first advantage is that rules in the TURS model can be empirically regarded as *truly* unordered and hence independent from each other. Thus, deleting and/or editing one rule (that a domain expert dislikes) has little influence on other, potentially overlapping rules. In contrast, for rules with (implicit) orders obtained by other existing methods, editing or deleting one rule may cause "a chain

of effects" on how instances covered by other rules are modeled. Secondly, the TURS model reduces the workload for domain experts to find out which rules need to be edited, because comprehending a single rule in TURS does not require going over all other (explicitly or implicitly) higher-ranked rules.

## 4 UPDATING RULE SETS WITH HUMAN FEEDBACK

We now describe in what format we allow ICU physicians to give feedback, and how the TURS model can be updated based on it.

### 4.1 Human feedback format

Although it seems tempting to allow feedback in flexible formats (like in natural language), we argue that it is desirable to constrain human feedback to have certain formats, in order to transform the feedback into *transparent human guidance* to the algorithm for updating the model. In other words, we aim to propose certain human feedback formats so that the consequence of such human feedback can be easily explained to domain experts.

However, such feedback format should also allow domain experts to express clearly and sufficiently why they dislike the current model. This requires a deep understanding about what might cause dissatisfaction from domain experts. Hence, how to design such feedback formats may depend on the application task at hand, and often require interdisciplinary collaborations.

Focusing on the task of ICU readmission risk analysis, we constrain ourselves to a simple yet fundamental feedback format and leave as future work incorporating other feedback formats. Formally, given a truly unordered rule set model with $K$ rules denoted as $M = \{S_1, ..., S_K\}$, we consider feedback from domain experts in the following form: *remove rule $S_j$ due to irrelevant variables $\{X_i\}_{i \in I}$*, in which $S_j$ denotes a single rule and $I$ an index set. Notably, feedback in this format contains both the information of whether a rule is disliked and the reason why it is disliked.

### 4.2 Updating a rule set

We now present how we can equip the TURS model with an "self-updating" scheme after receiving feedback from a domain expert.

**Removing a rule.** Given the rule set $M = \{S_1, ..., S_K\}$, assume that a domain expert gives the feedback that rule $S_i$ does not make sense as it contains irrelevant variable $X_j$. Then, removing $S_i$ from $M$ is straightforward as there exist no implicit or explicit orders among rules. That is, by following the procedure of formalizing a rule set as a probabilistic model [21], we can define a new rule set $M' = M \setminus \{S_i\}$, for which the likelihood can be calculated.

**Learn a new rule with constraint.** Building upon the new TURS model $M'$, we can learn a new rule by treating $M'$ by searching for the next "best" rule that optimizes the model selection criterion of TURS, with the constraint that the feature variable marked as "dislike" by domain experts will be skipped. The algorithm for searching the next rule is the same as in the original TURS algorithm, which adopts a beam-search approach [21].

## 5 AN EMPIRICAL PILOT STUDY

We conduct a pilot study in collaboration with Leiden University Medical Center (LUMC) using the real-world ICU patient dataset to showcase how the TURS model together with our proposed model updating scheme can be used for interactive rule learning with humans in the loop. We next describe the experiment setup and present our results.

### 5.1 Experiment setup

**Dataset description.** We specifically considered the dataset collected at the ICU of LUMC in the year 2020, in which the patients who are readmitted within 7 days are labelled as "positive".

The original dataset is multi-modal and contains information in different forms, including time series measurements (e.g., cardiology monitor records), lab results over time (e.g., blood tests), medication use records, as well as static information for each patient (e.g., age, gender, etc). This dataset was described and pre-processed into a tabular dataset by an external expert in previous research [16]. The resulting processed dataset was further split randomly for training and test, which contains 9737 and 2435 patients respectively (approximately 80%/20% splitting), with 550 feature variables. The dataset is very imbalanced, as the overall probability of readmission is roughly 0.07.

**Human feedback collection.** We ask one domain expert from LUMC to give feedback to the rules, with the procedure as follows. First, a TURS model is learned on the training set. Second, the rule set is shown to the domain expert; specifically, the condition of each rule together with the class probability estimates (obtained using the training set) are shown to the domain expert. Moreover, the algorithm is briefly described to the domain expert as well.

Next, we ask the domain expert to go through each of all rules, and to give feedback to the rule set in the format as we described in Section 4. Subsequently, the feedback is used to update the TURS model, and we use the test set of the ICU dataset for assessing the predictive performance of the TURS model before and after the human feedback. We refer to the latter as the human-guided model. Lastly, note that the test set of the whole dataset is only used for this final assessment step, and therefore the domain expert has no access to it during the procedure of giving feedback to rules.

### 5.2 Rule set for the ICU dataset

Learning a TURS model using the ICU dataset, we obtain a surprisingly simple rule set with 5 rules only, which has average rule length of 2. The obtained rule set is shown in Table 1.

The literals contain feature names that are mostly consisting of three parts, with the first part indicating the basic meaning of this feature variable. The second part of feature names indicates how the results are aggregated, among which "count", "mean", "median", and "max" are commonly used. Last, the third part of feature names indicates the time window for which the aggregated values are obtained, in which "first" represents the first 24 hours, "last" represents the last 24 hours, and "all" represents the whole period in ICU. A detailed explanation of the feature names can be found in a previous research [16].

**Table 1: Rule sets describing the probability of readmission for LUMC ICU patients.**

| Rule Conditions | Prob. of Readmission | # Patients |
|---|---|---|
| Urea-max-all ≥ 12.1 RespiratoryRate-median-value-last24h ≥ 23.5 | 0.223 | 494 |
| APTT-max-all ≥ 43.1 Urea-mean-all ≥ 16.338 | 0.199 | 548 |
| Leukocytes-mean-last ≥ 20.81 | 0.162 | 464 |
| Potassium-count-first ≥ 6.0 specialty-Organization-value-sub-ICCTC = FALSE | 0.131 | 1979 |
| Platelets-count-first ≥ 2.0 Urea-last-last < 9.2 specialty-Organization-value-sub-ICCTC = TRUE | 0.019 | 3922 |
| None of the above | 0.059 | 3220 |

## 5.3 Rule-based competitor methods

As a sanity check, we benchmark the performance of the TURS model induced from the training dataset against several commonly used probabilistic rule-based models. The motivation for such benchmark is to show that the TURS model has competitive predictive performance and thus implicitly describes the data relatively well, which is the foundation for involving humans in the loop.

The predictive performance is summarized in Table 2. Notably, the TURS model shows advantages over competitor methods in several aspects. First, the results with respect to ROC-AUC and PR-AUC show that the ICU dataset is difficult to model using widely used rule-based models (as listed in the table), since the ROC-AUC of C4.5 and RIPPER are roughly equal to 0.5. Further, the TURS model shows its robustness in achieving the best ROC-AUC and PR-AUC, and notably with significantly simpler rules (except when compared to RIPPER, which seriously "underfits" the data).

Moreover, rules in the TURS model generalize best to the unseen instances in the test set (excluding RIPPER for its low ROC-AUC scores). Specifically, we calculate the difference between the class probability estimates obtained using the training and test dataset, as also reported in the table. We hence conclude that the probability estimate for each single rule of the TURS model shown to physicians are most reliable and trustworthy.

**Table 2: Rule-based model results on ICU dataset.**

| Algorithm | CN2 | CART | RIPPER | C4.5 | TURS |
|---|---|---|---|---|---|
| ROC-AUC | 0.641 | 0.690 | 0.514 | 0.539 | 0.705 |
| PR-AUC | 0.114 | 0.137 | 0.084 | 0.076 | 0.164 |
| Train/test prob. diff. | 0.041 | 0.031 | 0.001 | 0.054 | 0.006 |
| # rules | 851 | 25 | 1 | 249 | 5 |
| Avg. rule length | 2.5 | 4.2 | 5.0 | 16.8 | 2.0 |

## 5.4 Human-AI collaboration

We now showcase that our TURS model can be equipped with the model updating scheme to generate human-guided rule sets. Notably, our approach is very different than existing model editing approaches [17], as the end user is not allowed to directly edit the model in our model updating scheme; instead, we only allow user to provide feedback, and the updated model is still learned in a data-driven manner. That is, we let the data always take the leading role, in order to avoid arbitrary (or adversarial) model editing.

We consider the rule set obtained in Section 5.2, and we collected two pieces of feedback from the domain expert: 1) the domain expert dislikes the 5th rule due to the first variable, and 2) the domain expert dislikes the 3rd rule which contains only one literal.

We thus discard the 5th rule from the rule set, and we next search for a new rule to be added to the rule set, with the constraint that the first variable in the 5th rule must not be included. We present the new human-guided rule together with the original rule in Table 3. We show that our TURS model indeed makes such an interactive process possible, and specifically that it can handle feedback that can be transformed into constraints with respect to excluding certain variables. Further, we demonstrate that for the rule set induced from ICU patients' dataset, editing a rule based on the human feedback (without the necessity to modify other 'overlapping' rules), can indeed discard certain variables but at the same time keep the predictive performance at the same level.

Note that the updated rule and the original rule are coincidentally very similar; that is, the feedback to the TURS model is only about discarding the first literal of the 5th rule, without asking it to keep the other literals and/or variables in the original rule.

**Table 3: Comparison between the rule before and after a domain expert feedback, together with the ROC-AUC and PR-AUC of the resulting new rule set. Changes in rules conditions before and after human feedback are shown in red and blue respectively.**

| Whether human-guided | No | Yes |
|---|---|---|
| Rule | If Platelets-count-first ≥ 2.0; Urea-last-last < 9.2; specialty-Organization-value-sub-ICCTC = TRUE → Probability of Readmission: 0.019; number of patients 3922 | If Leukocytes-count-first ≥ 2.0; Urea-last-last < 9.2; specialty-Organization-value-sub-ICCTC = TRUE → Probability of Readmission: 0.019; number of patients 3958 |
| ROC-AUC | 0.705 | 0.706 |
| PR-AUC | 0.164 | 0.164 |

Next, for examining the effect of the second feedback, we remove the 3rd rule from the original purely data-driven rule set, and search for another rule by excluding the variable "Leukocytes-mean-last" from the search space. We present the results in Table 4, which shows that the new rule covers 375 more patients than the original rule. Again, without the need for further modifying other rules, editing the 3rd rule in the original rule set with the updated rule keeps the ROC-AUC and PR-AUC at the same level.

## 6 CONCLUSION AND DISCUSSION

We studied the problem of estimating readmission risk for patients in ICU as an applied machine learning task. To resolve the difficult situation when domain experts (physicians) dislike certain rules, which can result in the lack of trust for such data-driven models,

**Table 4: Another comparison between the rule before and after a domain expert feedback.**

| Whether human-guided | No | Yes |
|---|---|---|
| Rule | Leukocytes-mean-last $\geq$ 20.8 $\rightarrow$ Probability of Readmission: 0.162; number of patients 464 | CRP-mean-last-missing = 1 $\rightarrow$ Probability of Readmission: 0.030; number of patients 839 |
| ROC-AUC (rule set) | 0.705 | 0.704 |
| PR-AUC (rule set) | 0.164 | 0.172 |

we developed a human-guided rule learning scheme based on our method for learning truly unordered rule set (TURS) models.

We presented a pilot empirical study using the patients data collected at Leiden University Medical Center (LUMC) in the year 2020. Specifically, we firstly presented the learned rule set from the ICU dataset, and compared the predictive performance against other widely used rule-based competitor models, which demonstrated the superiority of the TURS model in terms of both predictive performance and model complexity. This result set the foundation for using the TURS model as a basis for interactive rule learning.

Next, we asked a domain expert from LUMC to give feedback to the purely data-driven rules, and we proposed a simple model updating scheme to incorporate the feedback to induce human-guided rules. We showcased that such a process can lead to new rules as replacements for rules that the domain expert disliked, without sacrificing the predictive performance of the whole model. Notably, the properties of the TURS model enables straightforward, transparent, and efficient model editing, without the need for re-training other rules in the model. We next discuss potential future research directions.

## 6.1 Discussion for future work

We next discuss the following potential research directions.

**User feedback formats.** One natural question is in what formats we allow domain experts to give feedback to the data-driven model, and further how to inspire and elicit feedback with tools that allow an end user to investigate the data and the rule-based models.

For instance, it may be beneficial to allow domain experts to examine values of other features that are not included in the conditions of rules. While all instances in each rule share the same class probability estimate, domain experts may find one single "typical" patient who should have a different probability estimate than the rest. This may induce feedback in the form of "modifying a given rule by excluding a certain instance from the subset of instances covered by that rule".

Further, we could allow the domain experts to suggest informative feature to be included in a single rule. Thus, we may allow feedback in the form of "for all patients covered by this rule, those patients whose feature value for variable $X_i$ is larger than a certain threshold may have a higher risk of readmission". Such feedback is useful for 1) obtaining single rules with variables that are congruent

with the domain knowledge, and 2) more interestingly, understanding the limits of the data (since the "best" rule with the suggested variables may result in a "worse" score according to the model selection criterion).

**Transparent model updating.** Introducing the human in the loop extends the meaning of transparency of a machine learning method. Previously, transparency roughly referred to whether the process of obtaining a model based on a given dataset is comprehensible to humans; in contrast, we argue that transparency is also applicable to describing whether the process of model updating based on human feedback is comprehensible to humans. Thus, it is a natural question to ask whether the trust between domain experts and data-driven models depends not only on the transparency of the model but also on that of the model updating scheme.

Further, while it is very transparent to incorporate human feedback as constraints like those we proposed, other ways of processing human feedback are to be explored, e.g., translating human feedback to "prior" preferences.

## ACKNOWLEDGMENTS

We are profoundly grateful to Siri van der Meijden and Prof. Dr. Sesmu Arbous from Leiden University Medical Center for their unwavering support.

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
