# OpenReview forum: "Human-guided Rule Learning for ICU Readmission Risk Analysis"
_KDD.org/2024/Workshop/AIDSH — KDD-AIDSH 2024 Poster_

### Official Review · Reviewer_YdMz · 2024-06-14
**Review for Paper #13**

**Rating:** 5
**Confidence:** 4

**Review:**

## Summary

Building upon the Truly Unordered Rule Sets (TURS) model, the paper presents a method for predicting ICU readmission risks using an interactive machine learning approach that incorporates human feedback. The method allows for human-guided updates by ICU physicians. The feedback focuses on eliminating clinically irrelevant conditions in the rules. An empirical pilot study demonstrates that the updated rules maintain predictive performance while enhancing trust among physicians.

## Pros

- The background and method design are sound, as it is built upon an accepted ECML-PKDD paper: Truly unordered probabilistic rule sets for multi-class classification (TURS).
- The performance is promising, and it is evaluated on a real-world dataset by domain experts from the hospital.
- The human-guided ML system is indeed one of the directions to enhance physicians' trust in healthcare AI applications.

## Cons

- It would be better to include ML/DL-based baseline methods to see whether your approach exceeds those black-box AI models.
- Could you please provide the dataset statistics, including the proportion of readmission labels 1 and 0, and dataset size?
- Why is only one expert selected to evaluate and conduct the experiment?
- There should be train/val/test sets; the paper lacks the use of a validation set.
- The feedback format is unclear; the paper should include an illustrative figure.
- More metrics should be provided, like F1 scores.
- Only one task (readmission) is evaluated. Is it possible to provide mortality prediction results?

---

### Official Review · Reviewer_4jxb · 2024-06-15
**Review for Paper #13**

**Rating:** 7
**Confidence:** 5

**Review:**

Summary

This paper proposes a human feedback integration scheme based on the Truly Unordered Rule Sets (TURS) model for rule learning in the task of ICU readmission risk analysis. The authors first review the probabilistic rules and the TURS model, and then propose a human-guided rule update scheme based on the TURS model. Specifically, the authors present the rule set model to human users and ask them which rules they dislike and why. Then, the authors propose a transparent human-guided model update scheme and conduct an empirical study to demonstrate how to integrate human feedback into the TURS model. The results show that the rule learning process guided by human feedback can exclude variables considered irrelevant by experts while maintaining the predictive performance of the model.

Advantages:

1. The paper proposes a novel method of integrating human feedback into the TURS model to achieve automatic model update.
2. Through cooperation with the Leiden University Medical Center (LUMC), the paper conducts an empirical study using a real ICU patient dataset, demonstrating the impact of human feedback on the update of the TURS model and rule learning.
3. The model update scheme proposed in the paper is transparent because the learning of new rules is based on the feedback of domain experts as constraints.

Disadvantages:

1. The feedback format proposed in the paper is relatively simple, only allowing experts to point out disliked rules and the clinically irrelevant variables they contain. This may limit experts from providing more complex or in-depth feedback.
2. The paper only conducts experiments on the ICU readmission risk task and only uses one dataset, making it difficult to verify the generalization ability of the proposed method.
3. The comparison methods such as CN2 and CART in the paper are not introduced or referenced. Readers cannot intuitively understand whether these methods are competitive.

---

### Decision · Program_Chairs · 2024-06-28

Accept (Poster)